# Role of Histamine H_3_ Receptor Antagonists on Intraocular Pressure Reduction in Rabbit Models of Transient Ocular Hypertension and Glaucoma

**DOI:** 10.3390/ijms20040981

**Published:** 2019-02-24

**Authors:** Cecilia Lanzi, Laura Lucarini, Mariaconcetta Durante, Silvia Sgambellone, Alessandro Pini, Stefano Catarinicchia, Dorota Łażewska, Katarzyna Kieć-Kononowicz, Holger Stark, Emanuela Masini

**Affiliations:** 1Department of Neuroscience, Psychiatry, Drug Area and Child Health (NEUROFARBA), Section of Pharmacology and Toxicology, University of Florence, 50130 Florence, Italy; laura.lucarini@unifi.it (L.L.); mariaconcetta.durante@unifi.it (M.D.); sgambellonesilvia@gmail.com (S.S.); emanuela.masini@unifi.it (E.M.); 2Department of Experimental and Clinical Medicine, Section of Anatomy and Histology, University of Florence, 50139 Florence, Italy; alessandro.pini@unifi.it (A.P.); Stefano.catarinicchia@unifi.it (S.C.); 3Department of Technology and Biotechnology of Drugs, Jagiellonian University Medical College, 30-688 Krakow, Poland; dlazewska@cm-uj.krakow.pl (D.Ł.); katarzyna.kiec-kononowicz@uj.edu.pl (K.K.-K.); 4Institute of Pharmaceutical and Medicinal Chemistry, Heinrich Heine University Düsseldorf, 40225 Düsseldorf, Germany; stark@hhu.de

**Keywords:** histamine, intraocular pressure (IOP), glaucoma, oxidative stress, baroprotection, H_3_R antagonists

## Abstract

Intraocular pressure (IOP) has a tendency to fluctuate throughout the day, reaching its peak in the early morning in healthy subjects or glaucoma patients. Likewise, histamine tone also fluctuates over time, being lower at nighttime. Numerous studies have demonstrated a correlation between short-term IOP fluctuation and glaucoma progression; however, it has not yet been determined whether histamine plays a role in IOP fluctuations. The aim of this research was to establish the distribution of the histamine receptor proteins and respective mRNAs in the eye by western blot, immunohistochemistry and RT-PCR in New Zealand rabbits. Furthermore, we used a transient ocular hypertension (OHT) model induced by injection of 50 µL of 5% hypertonic saline into the vitreous and a stable OHT model (100 µL 0.1% carbomer in the anterior chamber) to address the potential IOP-lowering ability of H_3_ receptor (H_3_R) antagonists (ciproxifan, DL76 and GSK189254). IOPs were performed with a Tono-Pen at baseline and 60, 120 and 240 min post treatment after transient OHT induction and, every day for 12 days in the stable OHT model. All histamine receptor subtypes were localized in the rabbit retina and ciliary body/trabecular meshwork. All the treatments lowered IOP in a dose-dependent fashion between 0.3% and 1%. More specifically, the effects were maximal with ciproxifan at 60 min post-dose (IOP_60_ change = −18.84 ± 4.85 mmHg, at 1%), remained stable until 120 min (IOP_120_ change = −16.38 ± 3.8 mmHg, at 1%) and decayed thereafter to reach baseline values at 240 min. These effects were highly specific and dependent on histamine release as pre-treatment with imetit (H_3_R agonist, 1%) or pyrilamine (H_1_R antagonist, 1%) largely blocked ciproxifan-mediated effects. Color Doppler ultrasound examination was performed to evaluate changes in ophtalmic artery resistivity index (RI) before and after repeated dosing with DL 76, GSK189254, ciproxifan and timolol. Chronic treatments with H_3_R antagonists and timolol improved the vascular performance of ophthalmic arteries and reduced retinal ganglion cell death. Oxidative stress was also reduced and measured 8-Hydroxy-2′-deoxyguanosine (8OH*d*G) expression, and by dihidroethydium (DHE) staining. These results demonstrated that the histamine system participates in IOP regulation and that H_3_R antagonists could represent a future promising therapy for glaucoma. Further studies should be focused on the long-term IOP circadian fluctuations.

## 1. Introduction

Glaucoma is a multifactorial, degenerative disorder of the optic nerve. Progressive optic neuropathy afflicts more than 60 million people worldwide [1] and is mainly caused by glaucoma. Glaucoma is a group of eye diseases resulting in progressive ocular neuropathy, characterized by loss of retinal ganglion cells (RGC) and morphological alterations of the optic nerve head. It is mainly caused by excessively high intraocular pressure (IOP). This increased pressure within the eye, if untreated, can lead to a reduction of local perfusion, chronic ischemia leading to the reduction of RGC by autophagy, and apoptosis [2]. When the number of RGC is no longer suitable for neuronal transmission, the visual field becomes narrower. The first line of treatment consists of topical IOP-lowering medications, in other words a baroprotective strategy. A significant number of patients require more than one medication to achieve adequate baroprotection [3]. The target IOP reduction of 25% is not easily reached and maintained, especially at night. Detection of the progression of glaucoma is very important for treatment and intraocular pressure fluctuation is an important topic in recent works as a risk factor for glaucoma progression. Furthermore, long-term IOP circadian fluctuation is associated with a poor prognosis [4]. The posterior hypothalamus is involved in circadian rhythm regulation and histamine is one of the neuro-hormones implicated in the arousal system. Histamine tone is reduced at night and nocturnal IOP is higher than diurnal pressure; IOP is the result of a balance between secretion and outflow of aqueous humor. This balance is maintained perfectly in healthy subjects [5], with three variables being of interest: the rate of aqueous humor formation, resistance to outflow, and episcleral venous pressure. These variables change at night, with aqueous humor production decreasing during the night in diurnal mammals like rabbits while drainage facility decreases as well and, as a result, IOP increases [6]. The histamine system is deeply implicated in circadian rhythm and fulfills a major role in the maintenance of waking [7]. During the day, histamine tone could play a role in maintaining the IOP in balance. The histamine neurons are located exclusively in the posterior hypothalamus from which they project to most areas of the central nervous system and also to the retina via retinopetal axons [8]. Histamine is responsible for ciliary muscle contraction in human eyes and, therefore, for IOP reduction [9]. Taken together, this evidence supports a potential role of histamine in the regulation of IOP. Aqueous humor production or outflow rate can possibly be influenced by histamine, which is the neurotransmitter of the ascending arousal system, and the contributions of retinopetal axons to vision may be predicted from the known effects of histamine on retinal neurons [8].

This research aims to validate H_3_ receptor antagonists, both imidazole (ciproxifan) and non-imidazole compounds (DL76 and GSK189254) in IOP reduction, and establish whether the histamine system plays a role in baroprotection and, consequently, in retinal neuroprotection by reducing oxidative stress. The project also investigates the presence and distribution of histamine receptor subtypes in the retina, ciliary bodies and trabecular meshwork (TM).

## 2. Results

### 2.1. Histamine Receptor Protein and mRNA Expression in the Eye of New Zealand White Rabbits

The protein expression of H_1,_ H_3_ and H_4_ receptors was found in the retina, optic nerve and TM. H_2_ receptor (H_2_R) was undetectable in the retina, optic nerve and TM (Figure 1A,B). The mRNA expression of histamine receptors was present in the retina and TM (Figure 1C). The hippocampus was used as a positive control for H_1_ and H_3_ receptors, the stomach for H_2_R and the spleen for the H_4_R. The mRNA of the H_2_R was found in all the tissues studied, while expression of the protein was not. Immunofluorescence analysis revealed the presence of histamine H, H_3_ and H_4_ receptors in the RGC (Figure 1D).

The presence of mRNA and protein expression of histamine H_3_R led us to investigate the effects of different H_3_R antagonists (GSK189254, ciproxifan and DL 76) on IOP reduction in models of ocular hypertension.

### 2.2. Pharmacological Studies of H_3_R Antagonists in Transient Ocular Hypertension Model

Selection of the best dose of H_3_R antagonists was carried out in different experimental sets using the transient ocular hypertension model. In the ciproxifan experimental set, IOP rose from 16.8 ± 5.6 mmHg at baseline to 39.63 ± 4.85 mmHg after hypertonic saline injection (Figure 2A). Reduction of IOP was greatest with ciproxifan (IOP_60_ change = −18.84 ± 4.85 mmHg, at 1%): remaining stable until 120 min (IOP_120_ change −16.38 ± 3.8 mmHg, at 1%) and decaying thereafter to reach baseline values at 240 min (Figure 2B). In the DL76 experimental set, IOP rose from 16.5 ± 3.7 mmHg at baseline to 39.5 ± 5.2 mmHg after hypertonic saline injection (Figure 2C), IOP_60_ change at 1% was −17.45 ± 4.48 mmHg and remained stable until 120 min (IOP_120_ change −18.38 ± 3.04 mmHg, at 1%); in the GSK189254 experimental set, IOP rose from 14.9 ± 4.2 mmHg at baseline to 40.2 ± 4 mmHg (Figure 2E). The IOP_60_ change at 1% was −8.61 ± 4.18 mmHg and the IOP_120_ change was −9.92 ± 9.02 mmHg. All the compounds reduced IOP dose-dependently and in a statistically significant manner with a different profile, ciproxifan and DL76 being more effective than GSK189254 (Figure 2B,D,F). After these series of experiments, we compared the three H_3_R antagonists at 1% dose with the gold standard treatment timolol at 1% dose. In this experimental group, IOP rose from 15.7 ± 3.4 mmHg at baseline to 37.7 ± 4.2 mmHg after hypertonic saline injection (Figure 2G); ciproxifan and DL76 showed an IOP-lowering profile very similar to timolol (timolol IOP_60_ change = −16.5 ± 2.6 mmHg and IOP_120_ change = −15.12 ± 2.85 mmHg in Figure 2H). No adverse side effects were observed and the drugs did not cause any changes in pupil diameter.

### 2.3. Evaluation of Specificity of H_3_R Antagonistic Action

We evaluated the receptor specificity on IOP reduction by counteracting the H_3_R antagonism with a pretreatment with 1% imetit, a histamine H_3_R agonist. IOP rose from 16.2 ± 3.5 mmHg at baseline to 40.2 ± 5.3 mmHg after hypertonic saline injection (Figure 3A). The IOP-lowering activity of ciproxifan was suppressed by imetit pre-treatment (Figure 3B). Treatment with 1% pyrilamine, a H_1_R antagonist or with 1% amthamine, a H_2_R agonist, was also performed. In this experimental set, IOP rose from 18.8 ± 4.2 mmHg at baseline to 40.2 ± 3.5 mmHg (Figure 3C); The pretreatment with pyrilamine reverted the hypotensive action of ciproxifan (Figure 3D). Amthamine alone had no effect on IOP.

### 2.4. Evaluation of the IOP Profile after Choroidal Mast Cell Granule Depletion by Compound 48/80 Pre-Treatment

In this experimental set, IOP rose from 17.7 ± 3.1 mmHg at baseline to 38.8 ± 3.7 mmHg after hypertonic saline injection (Figure 4A). IOP-lowering activity of ciproxifan 1% was suppressed in animals pretreated with compound 48/80 (C48/80) for two days before the IOP profile experiments (Figure 4B). Ex vivo choroid examination revealed an intense and statistically significant mast cell degranulation (Figure 4C,D).

### 2.5. Effects of Histamine H_3_R Antagonists on IOP, Ocular Vascular Dynamics, Retinal Ganglion Cell Conservation and Oxidative Stress in Glaucoma Model

To better understand the role of H_3_ antagonism on modulating IOP, we evaluated both non-imidazole (GSK189254 and DL76) and imidazole (ciproxifan) compounds in a stable ocular hypertension model using timolol as a reference drug, with a daily dosing schedule. IOP rose from 13.5 ± 3.8 to 38.3 ± 1 two days after carbomer injection and remained stable for two weeks in vehicle-treated animals (Figure 5A). Treatment with 1% ciproxifan and DL76 induced a statistically significant reduction of IOP starting from day 5. The effect was maintained throughout the whole experimental period (Figure 5B).

To evaluate the vascular performance at the posterior pole of the eye, we measured systolic and diastolic velocities by means of Doppler ultrasound studies of the ophthalmic arteries. The Pourcelot resistivity index (RI) significantly increased in glaucomatous animals compared to naïve ones. The treatments with 1% ciproxifan or 1% DL76 significantly reduced Pourcelot RI. Similar results were obtained with 1% timolol (Figure 5C).; mean RI values are reported in Figure 5D.

The death of RGC is the main cause of visual impairment in glaucoma. Repeated-dosing treatment with H_3_R antagonists, as well as with timolol, significantly prevented the death of neurons in the RGC layer. Morphological evaluation of retinae of ciproxifan and timolol groups showed a slightly better conservation of RGC (Figure 5E,F).

Oxidative stress was quantitatively evaluated by determining 8-Hydroxy-2′-deoxyguanosine (8OH*d*G), a marker of DNA damage and, morphologically with dihydroethidium (DHE) staining. Ciproxifan, DL76 and also timolol significantly reduced 8OH*d*G formation (Figure 5G). DHE staining demonstrated that the intensity of signal was very strong in the retina of vehicle-treated hypertensive eyes, showing a higher oxidative damage in the nuclei of cells of retinal layers of the vehicle group in comparison with those of the control (naïve). H_3_R antagonists and timolol-treated eyes presented retinal images with a less intense fluorescent signal, thus meaning less damage to DNA.

## 3. Discussion

Our results confirm that histamine plays an important role in IOP regulation. Histamine receptors are expressed in a neuronal and non-neuronal compartment in the eye in diurnal mammals like rabbits. In diurnal animals during the night, although aqueous humor production decreases, IOP increases because of reduced aqueous humor drainage from the eye [6] and this happens when the histaminergic system, the arousal system, is less active. In patients enrolled in the Advanced Glaucoma Intervention Study (AGIS), long-term IOP circadian fluctuation, i.e., ocular hypertension at night, was associated with disease progression [10]. We demonstrated that histamine receptors are expressed in the retina and TM, both as mRNA and protein levels, with the exception of H_2_R, which is expressed only as mRNA. Our results are in agreement with previous publications demonstrating histamine H_1_R in ciliary bodies, preferentially on endothelial cells [11], and H_3_R of ON-bipolar cell dendrites, in cone pedicles and rod spherules [12]. 

Topical treatments with H_3_R antagonists were effective in reducing IOP both in transient and stable OHT animal models. Treatments with imidazole (ciproxifan) and non-imidazole (DL76 and GSK189254) compounds were effective in decreasing IOP at 60 and 120 min more than 25% of the reduction target, in a dose-dependent way. GSK189254 was less effective in reducing IOP in our rabbit model. To explain it we need to know more about the p*K*i of this compound in rabbit tissue, as p*K*i in the rat brain was similar to those of the other H_3_ antagonists. GSK189254 is an H_3_R antagonist with high affinity for human (p*K*i = 9.59–9.90) and rat (p*K*i = 8.51–9.17) H_3_Rs comparable to those of ciproxifan and DL76 (Table A1).

The IOP lowering profile of ciproxifan was counteracted by pre-treatment with the H_3_R agonist imetit, confirming a H_3_R mechanism. The effect of pyrilamine in the prevention of the hypotensive effect of ciproxifan could implicate the involvement of the H_1_ receptor in the IOP lowering process. To confirm this hypothesis and to investigate the local source of histamine, we depleted mast cells (MC) in the choroid tissue of their granule content with compound 48/80 [13] and, as expected, we did not observe any ciproxifan effect, suggesting a role of MC histamine in the regulation of IOP. It is important to consider that, in the central nervous system, approximately 50% of histamine belongs to the non-neuronal pool [14]. The presence of MC in the meninges and the perivascular district of the brain blood barrier indicates that this cell type is relevant in neurovascular responses [15]. It is also possible that compound 48/80-mediated massive release of histamine could down-modulate the H_1_ receptors involved in regulation of the intraocular pressure, reproducing the effect of H_1_ blockade observed with pyrilamine pretreatment. Histamine could interact with the H_1_ receptor, a G-protein receptor (Gq), on the cell surface and activates phospholipase C and the phosphatidylinositol signal pathway, thus increasing intracellular calcium inducing cell contraction (Figure 6). Histamine, via H_1_R, stimulates the production of inositol phosphate and mobilization of intracellular Ca^2+^ in cultured human ciliary muscle cells, reducing IOP [9,16].

Many animal glaucoma models have been developed over the decades [17] and we chose the carbomer one for its simplicity and reproducibility in order to evaluate the onset and pathological progression of ocular hypertension in a controlled and reproducible manner [18,19]. The demonstration of histamine H_3_R subtype in the ciliary body and TM, not only in the retina, is an original result. The TM is a specialized tissue responsible for draining the majority of aqueous humor (AH) from the anterior chamber of the eye, thereby controlling IOP. The localization of H_3_R at this level is therefore particularly interesting.

Histamine H_3_Rs are not expressed exclusively in neurons and not only at a presynaptic level [20]. They are expressed on distinct endocrine cell types in the rat fundi mucosa [21], in the apical membrane, and near collecting duct cells in the rat kidney, and their expression is significantly increased in diabetic animals, suggesting its involvement in fluid homeostasis [22].

In the chronic treatment schedule in both ciproxifan and DL76 groups, we observed a decrease of IOP from the third day of topical treatment, which became statistically significant from day 5. This delay in the effect suggests a mechanism of action that involves a medium-term process in ocular pathology. The efficacy of the treatments and therefore an achieved baroprotection is confirmed by the observation that RGC, evaluated in hematoxylin/eosin-stained retinal sections and morphometrically quantified, were maintained in the eyes of animals that achieved an adequate and stable IOP reduction during the experimental procedures. Moreover, we also observed an important reduction of oxidative stress parameters. The presence of oxidative stress markers is one of the constant findings in the progression from OHT to optic neuropathy [23]. Oxidative stress markers in the retinae were significantly reduced by baroprotective treatments. This process was helped by H_3_ antagonist treatments that improved vascular performance of ophthalmic arteries, reducing the RI, acting on the third variable of interest in IOP balance. The TM is responsible for approximately 75% of outflow resistance [24], but the implementation of the vascular performance is important to maintain an adequate perfusion of the posterior pole of the eye [25]. Vascular impairment in the eye district is important in the pathogenesis of normotensive glaucoma. 

In conclusion, our findings demonstrate that the histamine system participates in IOP regulation. H_3_R antagonists significantly reducing ocular hypertension, preventing RGC loss by an improvement of vascular performance of the central ophthalmic artery, and reducing oxidative stress could represent a future promising therapy for glaucoma.

## 4. Materials and Methods

### 4.1. Animal Models of Glaucoma

The experimental procedures were carried out in New Zealand albino rabbits. All procedures on animals conformed to the Association for Research in Vision and Ophthalmology Resolution, in agreement with the Good Laboratory Practice for the use of animals, with the European Union Regulations (Directive 2010/63/EU), upon authorization of the National Ethics Committee of the Italian Ministry of Health (number 1179/2015-PR; 16 November 2015). Male albino rabbits (body weight 2–2.5 kg) were kept in individual cages; food and water were provided ad libitum. The animals were maintained on a 12–12 h light/dark cycle in a temperature-controlled room (22–23 °C). Animals were identified with a tattoo on the ear, numbered consecutively. All selected animals underwent ophthalmic and general examinations before the beginning of the study.

A total of 12 rabbits were used for the parallel study of each of the four compounds and for the vehicle. The probability of reaching a statistically significant difference between treatment and control groups was set at 80% with a significance level of 0.05 if the mean difference between treatments was 8 mmHg. The software used for this calculation was a free software called G* power 3 [26].

### 4.2. Rabbit Models of Ocular Hypertension and Glaucoma

The transient IOP elevation model (OHT model) was obtained by injection of 0.05 mL of sterile hypertonic saline (5%) into the vitreous bilaterally in locally anesthetized rabbits with one drop of 0.2% oxybuprocaine hydrochloride [27]. In this model, ocular hypertension was reached in 10 min and then slowly decreases for 4 h. Compounds were instilled into the lower conjunctival pocket. All the compounds were given prior to saline injection, and IOP was measured at the very beginning of the experimental session to establish basal IOP and subsequently at 60, 120 and 240 min.

The chronic IOP elevation model (glaucoma model) was obtained by injection of 0.1 mL carbomer 0.25% (Siccafluid, Farmila THEA Pharmaceutical, Milan, Italy) bilaterally into the anterior chamber of New Zealand albino rabbits pre-anesthetized with zolazepam plus tiletamine (Zoletil 100, 0.05 mg/kg) plus xylazine (Xilor 2%, 0.05 mL/kg) i.m. [18,19]. In this model, ocular hypertension was reached in 48 h and maintained for at least two weeks. To evaluate the reliability of the model, IOP was measured three times a day (at 8 a.m., 2 and 8 p.m.) until stabilization for the first 48 h [28], and then measured each morning before drug instillation. The carbomer model is a two-week-long period of “once a day” eye drop instillation and tonometry evaluation.

### 4.3. IOP Measurements

All IOP measurements were performed by applanation tonometry using a tonopen Avia (Reichert Technologies, Depew, NY, USA). This method has proven to be as reliable as Goldmann applanation tonometry or GAT, the most accurate way of measuring IOP [29]. GAT is considered the gold standard for IOP measurement in clinical practice [30]. Tonopen Avia tonometry was used both in the transient and chronic IOP elevation model. One drop 0.2% oxybuprocaine hydrochloride diluted 1:1 with saline was instilled in each eye immediately before each set of pressure measurements. The change in IOP was calculated as the difference between the mean value of IOP in the different treatments towards the mean IOP in vehicles at the corresponding timepoints. 

### 4.4. Compounds Used in the Experiments 

All compounds were purchased by Sigma Aldrich (Milan, Italy) or Tocris Cookson Inc. (Bristol, UK) with the exception of ciproxifan, provided by Prof. Holger Stark (Heinrich Heine Institute, Dusseldorf University, Dusseldorf, Germany) and DL76, a novel non-imidazole H_3_R antagonist 1-[3-(4-tert-butylphenoxy)propyl]piperidine hydrogen oxalate provided by Dr. Dorota Łażewska and Prof. Katarzyna Kieć-Kononowicz (Department of Technology and Biotechnology of Medicinal Drugs, Jagiellonian University, Kraków, Poland) [31,32,33,34]. Histamine ligands are reported with their relative receptor affinities in Table A1 [35]. Compound 48/80 (C48/80) was instilled in the eye at a concentration of 15 µg/kg body weight diluted in saline. The Draize eye test was used to establish if the formulation of each test compound was suitable for ocular administration [36].

### 4.5. Pharmacological Evaluation of H_3_R Antagonists on IOP Reduction in Transient OHT Model: Dose/Effect and Specificity

The experimental design to find the best dose of different H_3_R antagonists was carried out in the transient IOP elevation model with the instillation of an eye drop of 30 µL of compounds at three different concentrations (0.1, 0.5, 1%) in the ocular cul de sac. The pharmacological evaluation of the H_3_R antagonism was studied by pre-treating the animals with 30 µL of the H_3_R agonist imetit at equimolar concentration with the H_3_R antagonist. The same procedure was carried out with the H_1_R antagonist pyrilamine and the H_2_R agonist amthamine.

### 4.6. Pharmacological Evaluation of H_3_R Antagonists on IOP Reduction and Baroprotection in Glaucoma Model

We tested the effects of ciproxifan, DL 76, GSK189254, and timolol maleate versus vehicle in reducing IOP measured as previously described. We evaluated the baroprotective effects of the different compounds by hemodynamic evaluation using an eco-color-doppler Philips Ultrasound HD7 XE (Brothel, WA, USA). All animals underwent color Doppler imaging (CDI) investigation at the beginning of the experimental procedure and at the end of drug treatments. Special attention was devoted to the evaluation of retinic artery circulation. Blood flow velocities were measured and the Pourcelot resistivity index (RI) calculated [19]. Pourcelot RI, a significant indicator of ocular vascular impairment, is the ratio given by the difference between systolic and diastolic velocity over systolic velocity.

### 4.7. Sample Collection

Collection of retinae and ciliary bodies was performed in the basal condition and at the end of experimental glaucoma. Animals were killed with a lethal dose of anesthetics (zolazepam/tiletamine and penthotal sodium 0.05g/kg). For each treatment group at least three eyes were excised and fixed with 4% paraformaldehyde in phosphate-buffered saline for histological analysis and the contralateral ones dissected to collect the retina and ciliary body, quickly frozen in liquid nitrogen and stored at −80°C. Samples of stomach, spleen, and hippocampus were also collected to use as positive controls for H_2_, H_4_ and H_1_/H_3_R, respectively.

### 4.8. Western Blot Analysis of Histamine Receptors

Ciliary bodies, retinae, and stomach were homogenized in ice-cold lysis buffer (NaCl 0.9%, tris-HCl 20 mmol/L pH 7.6, Triton X-100 0.1%, phenylmethylsulfonyl fluoride 1 mmol/L, leupeptin 0.01%) plus a protease inhibitor cocktail (Cell Signaling, Milan, Italy), sonicated, and centrifuged at 4 °C for 10 min at 10,000 *g*. Total protein levels were quantified using the Pierce™ BCA Protein Assay Kitc (Thermo Fisher Scientific, Rockford, IL, USA). Tissue samples (30 μg protein per lane) were subjected to 8% SDS-PAGE, transferred to nitrocellulose membranes and incubated overnight (4 °C) with H_1_, H_2_, H_3_ and H_4_R antibodies (Santa Cruz Biotechnology, Dallas, TX, USA). Primary antibodies were diluted 1:1000 in T-PBS (20 mM Tris-HCl buffer, 150 mM NaCl and 0.05% Tween 20) containing 5% BSA. After three rinses with T-PBS, membranes were incubated with secondary antibodies IgG conjugated with peroxidase, diluted 1:5000 in T-PBS at RT for 1 h. β-actin protein was used as the internal control for normalization. The immunoreaction was revealed by enhanced chemiluminescence (Luminata Crescendo Western HRP substrate, Merck Millipore, Darmstadt, Germany) and quantified by densitometric analysis with the Image J software (version 1.50, National Institutes of Health, Bethesda, MD, USA)

### 4.9. Semi-Quantitative RT-PCR of Histamine Receptors

Total RNA was isolated from a tissue according to the manufacturer’s protocol (NucleoSpin RNA II; Machery-Nagel, Bethlehem, PA, USA). Concentration and purity were assessed by spectrophotometric analysis (Genequant-Pro, GE Healthcare, Milan, Italy). One μg of total RNA was reverse-transcribed to cDNA with M-MLV Reverse Transcriptase (Omniscript; QIAGEN, Milan, Italy) using random primers in a 20 μL reaction incubated at 25 °C for 5 min followed by 60 min at 42 °C and 5 min at 70 °C. Two hundred ng of cDNA was amplified as follows: 15 min at 95 °C, 1 min of denaturation at 94 °C, 30 s of annealing, and 30 s of extension at 72 °C for 38 cycles. Primer sequences are summarized in Table 1. Amplification products were highlighted with ethidium bromide on 1.5% agarose gel. The intensities of the bands were quantified by densitometric analysis. The used histamine receptor gene sequences are reported in Table 1.

### 4.10. Immunofluorescence of Ciliary Bodies and Retinae

Histological sections, 5 μm thick, were cut from the paraffin-embedded posterior and anterior eye samples. The sections were deparaffinized and boiled for 10 min in sodium citrate buffer (10 mM, pH 6.0; Bio-Optica, Milan, Italy) for antigen retrieval. After a pre-incubation in 1.5% bovine serum albumin (BSA) in PBS, pH 7.4 for 20 min at RT to minimize the non-specific binding, the sections were incubated overnight at 4 °C with primary antibodies directed against H1-4, diluted 1:100 in 5% BSA PBS-T (Santa Cruz Biotechnology, Dallas, TX, USA), followed by secondary Alexa Fluor 594-conjugated IgG (1:500; Jackson ImmunoResearch Europe Ltd., Cambridge House, UK).

The sections were mounted in an aqueous medium (Fluoremount, Sigma, Milan, Italy) with 4′,6-diamidino-2-phenylindole (DAPI). Representative images were acquired by an Olympus BX63 microscope coupled to CellSens Dimension Imaging Software version 1.6 (Olympus, Milan, Italy).

### 4.11. Determination of 8-Hydroxy-2′-deoxyguanosine

Determination of 8-Hydroxy-2′-deoxyguanosine (8OH*d*G) was performed in order to evaluate the oxidative stress induced by chronic ocular hypertension. Frozen ciliary bodies and retinae samples were thawed at room temperature, and cell DNA isolation was performed as previously described with minor modifications [37]. Briefly, eye samples were homogenized in 1 mL of 10 mM PBS, pH 7.4, sonicated on ice for 1 min, added with 1 mL of 10 mmol/L Tris-HCl buffer, pH 8, containing 10 mmol/L EDTA, 10 mmol/L NaCl, and 0.5% SDS, incubated for 1 h at 37 °C with 20 μg/mL RNase 1 (Sigma-Aldrich, Saint Louis, MO, USA) and overnight at 37 °C under argon in the presence of 100 μg/mL proteinase K (Sigma-Aldrich, Milan, Italy). The mixture was extracted with chloroform/isoamyl alcohol (10/2 *v*/*v*). DNA was precipitated from the aqueous phase with 0.2 volumes of 10 mmol/L ammonium acetate, solubilized in 200 μL of 20 mmol/L acetate buffer, pH 5.3, and denatured at 90 °C for 3 min. The extract was then supplemented with 10 IU of P1 nuclease (Sigma-Aldrich) in 10 μL and incubated for 1 h at 37 °C with 5 IU of alkaline phosphatase (Sigma-Aldrich) in 0.4 mol/L phosphate buffer, pH 8.8. All of the procedures were performed in the dark under argon. The mixture was filtered by an Amicon Micropure-EZ filter (Merck-Millipore Darmstadt, Germany), and 50 μL of each sample was used for 8OH*d*G determination using an ELISA kit (JalCA, Shizuoka, Japan), following the instructions provided by the manufacturer. The absorbance of the chromogenic product was measured at 450 nm. The results were calculated from a standard curve based on an 8OH*d*G solution and expressed as ng/ng of DNA.

### 4.12. Fluorescent Dye Dihydroethidium (DHE) Staining

Histological sections, 5 µm thick, were cut from the paraffin-embedded posterior and anterior eye samples, deparaffinized and incubated with fluorescent DHE 2µM (Abcam, Camdridge, UK) in a humidity chamber for 30 min at 37 °C. Coverslips were then placed on the slides and kept in the dark. The sections were mounted in an aqueous medium (Flouremont, Sigma, Milan, Italy). Flourescence was detected (absorbance 518 nm, emission 605 nm) using an Olympus BX61 microscope coupled to CellSens Dimension Imaging Software, version 1.6 (Olympus, Milan, Italy) 

### 4.13. Statistical Analysis

For each assay, data were reported as mean values (± SEM) of individual average measures of the different animals per group. The significance of differences among the groups was assessed by one-way ANOVA or two-way ANOVA for multiple comparisons followed by the Bonferroni post hoc test and unpaired t-test with Welch’s correction. Calculations were made with Prism 5.1 statistical software (GraphPad Software Inc., San Diego, CA, USA). A probability value (*p*) of <0.05 was considered significant.

## Figures and Tables

**Figure 1 ijms-20-00981-f001:**
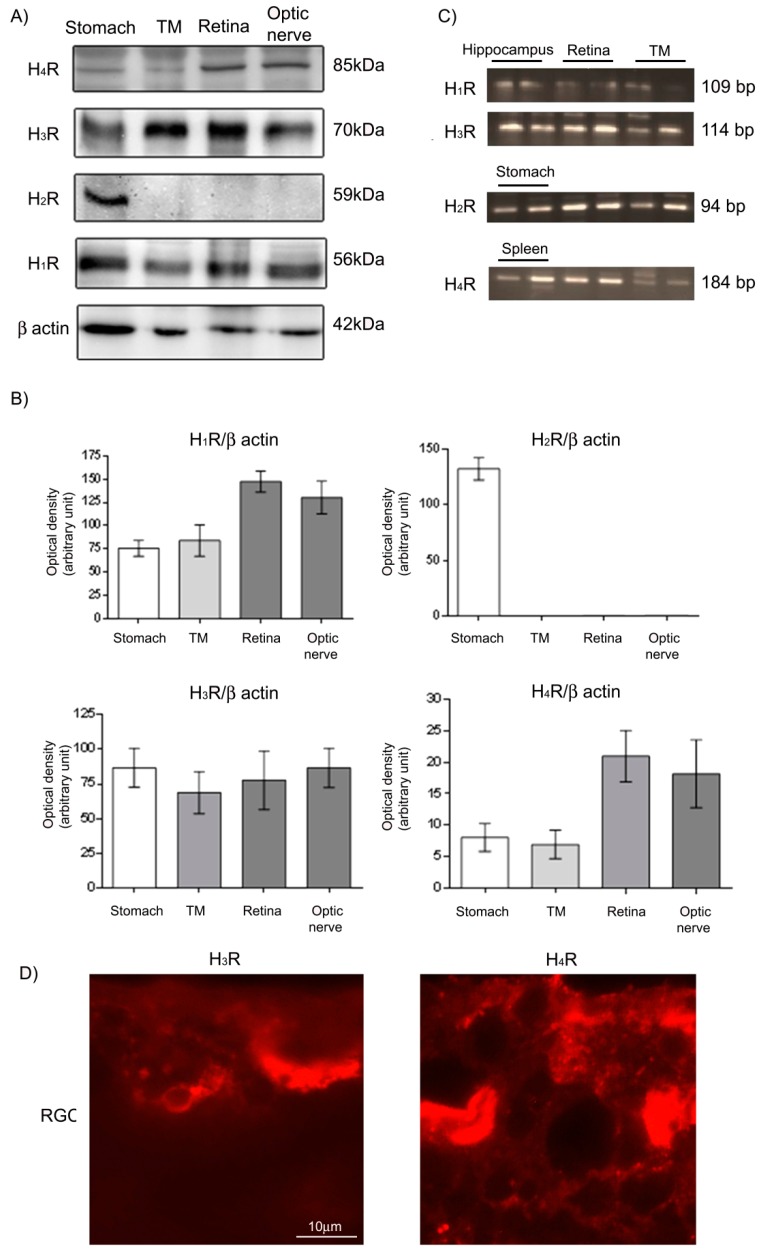
(**A**) Protein expression of histamine receptor subtypes by Western blot analysis; (**B**) optical densitometry of protein expression, *n* = 5; (**C**) mRNA expression by RT-PCR of histamine receptor subtypes in the trabecular meshwork (TM), retina and hippocampus, *n* = 5; (**D**) representative images of H_3_ and H_4_ receptors (H_3_R and H_4_R) in retinal ganglion cells (RGC) at 100*×* magnification.

**Figure 2 ijms-20-00981-f002:**
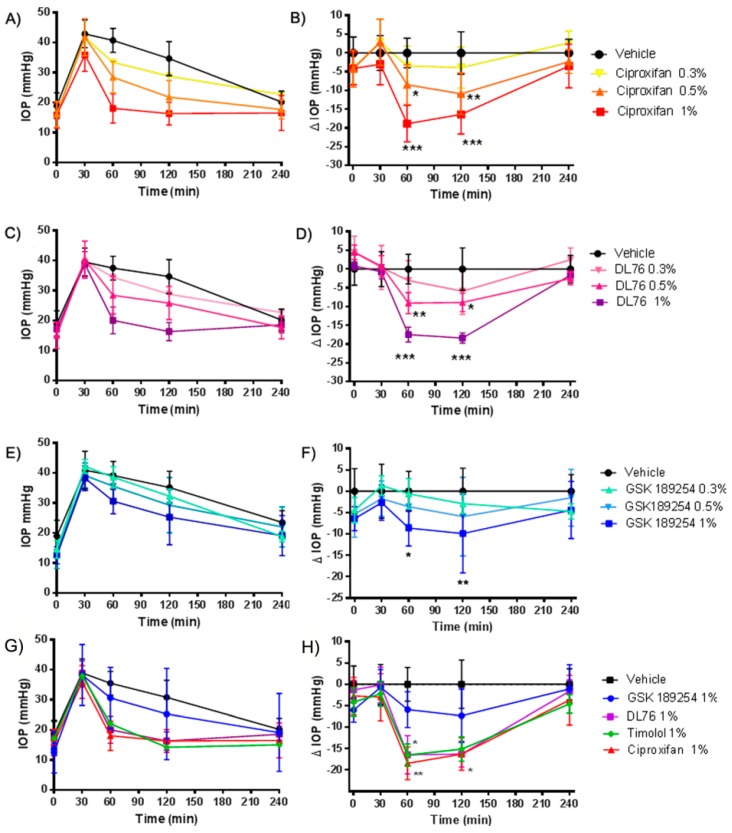
(**A**,**B**) Ciproxifan intraocular pressure (IOP) time course. *** *p* < 0.001 ciproxifan 1% at 60′ and 120′; ** *p* < 0.01 ciproxifan 0.5% at 120′; * *p* < 0.05 ciproxifan 0.5% at 60′ versus vehicle; (**C**,**D**) DL76 IOP time course. *** *p* < 0.001 DL76 1% at 60′ and 120′; ** *p* < 0.01 DL76 0.5% at 60′; * *p* < 0.05 DL76 0.5% at 120′ versus vehicle; (**E**,**F**) GSK189245 IOP time course. ** *p* < 0.01 GSK189254 1% at 120′; * *p* < 0.05 GSK189254 1% at 60′ versus vehicle; (**G**,**H**) effect of H_3_ antagonists versus timolol. ** *p* < 0.01 ciproxifan 1% at 60′; * *p* < 0.05 DL76 and timolol 1% at 60′ and 120′ versus vehicle. All the results are expressed as mean ± SEM (*n* = 5). Two-way ANOVA followed by Bonferroni post hoc test. Compounds were instilled in drops into the lower conjunctival pocket.

**Figure 3 ijms-20-00981-f003:**
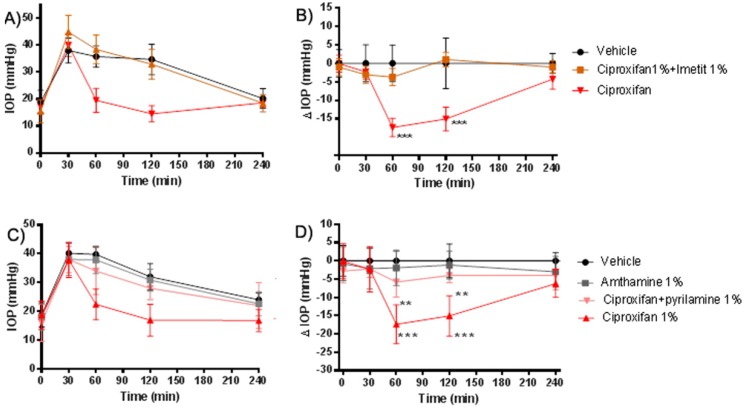
(**A**,**B**) Ciproxifan-Imetit intraocular pressure (IOP) time course. *** *p* < 0.001 ciproxifan 1% versus vehicle and versus ciproxifan + imetit 1% at 60′ and 120′; (**C**,**D**) amthamine and pyrilamine IOP time course. *** *p* < 0.001 ciproxifan 1% at 60′ and 120′ versus amthamine and vehicle; ** *p* < 0.01 ciproxifan versus ciproxifan + pyrilamine. Data are expressed as mean ± SEM (*n* = 5). Two-way ANOVA followed by Bonferroni post hoc test. Compounds were instilled in drops into the lower conjunctival pocket.

**Figure 4 ijms-20-00981-f004:**
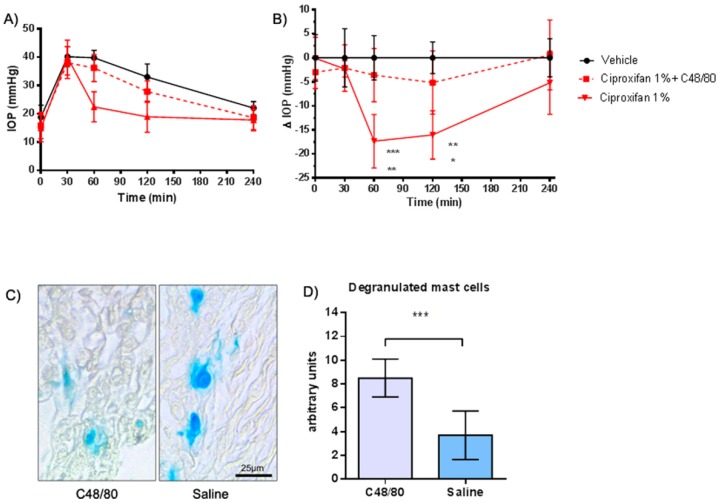
(**A**,**B**) C48/80 intraocular pressure (IOP) time course. *** *p* < 0.001 ciproxifan 1% at 60′ versus vehicle; ** *p* < 0.01 ciproxifan 1% at 60′ versus ciproxifan + C48/80, ciproxifan 1% versus vehicle at 120′; * *p* < 0.05 ciproxifan 1% at 60′ versus ciproxifan + C48/80. Data are expressed as mean ± SEM (*n* = 5). Two-way ANOVA followed by Bonferroni post hoc test.; (**C**) Blue Astra staining of mast cells in the choroid; (**D**) *** *p* < 0.001 C48/80 treated eyes versus saline-treated eyes with Welch’s correction *t*-test. Compounds were instilled in drops into the lower conjunctival pocket.

**Figure 5 ijms-20-00981-f005:**
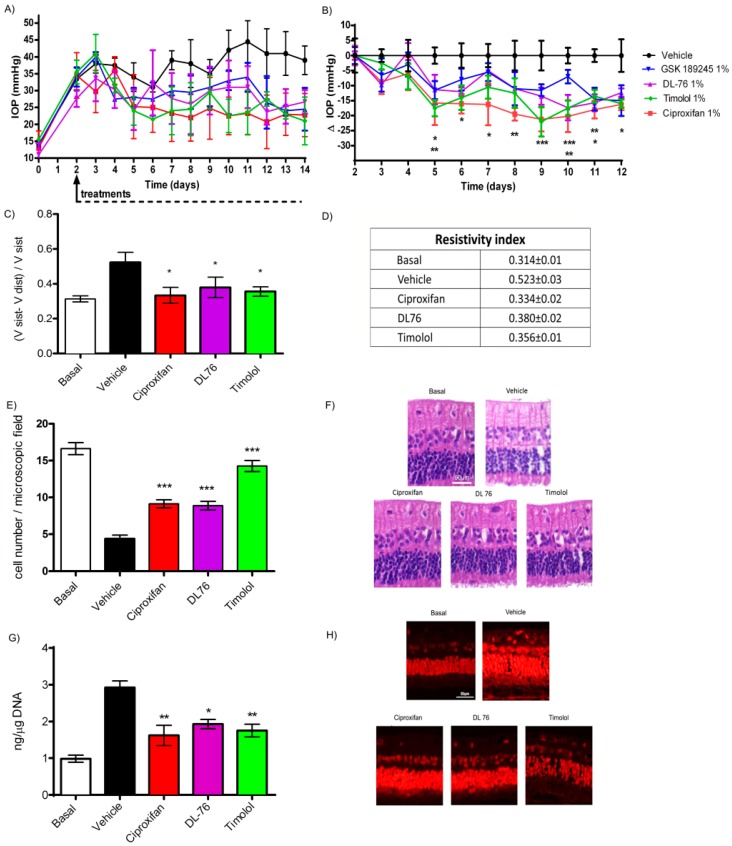
(**A**,**B**) Intraocular pressure (IOP) time course. *** *p* < 0.001 ciproxifan at day 9 and 10; ** *p* < 0.01 ciproxifan day 5, 8, 11 timolol and DL76 at day 10; * *p* < 0.05 ciproxifan at day 5, 6, 7, 12, DL76 and GSK 189254 at day 11, 12 and timolol at day 12 versus vehicle; (**C**,**D**) Pourcelot resistivity index. * *p* < 0.05 basal (naïve animal) versus vehicle; (**E**) Retinal ganglion cells (RGC) count: *** *p* < 0.001 Timolol versus vehicle (glaucomatous animal). *** *p* < 0.001 ciproxifan and DL76 a 1% versus vehicle. (*n* = 5); (**F**) representative images of hematoxylin/eosin-stained histological sections of retinae from different treated groups. RGC are visible in the upper layer; (**G**) 8-Hydroxy-2′-deoxyguanosine (8OH*d*G), levels. ** *p* < 0.01 ciproxifan and timolol 1%; * *p* < 0.05 DL76 versus vehicle; (**H**) representative images of DHE stained histological sections of retina. Data are expressed as mean ± SEM (*n* = 5). Two way (**B**) and one way (**C**, **E** and **G**) ANOVA followed by Bonferroni post hoc test. Compounds were instilled in drops into the lower conjunctival pocket.

**Figure 6 ijms-20-00981-f006:**
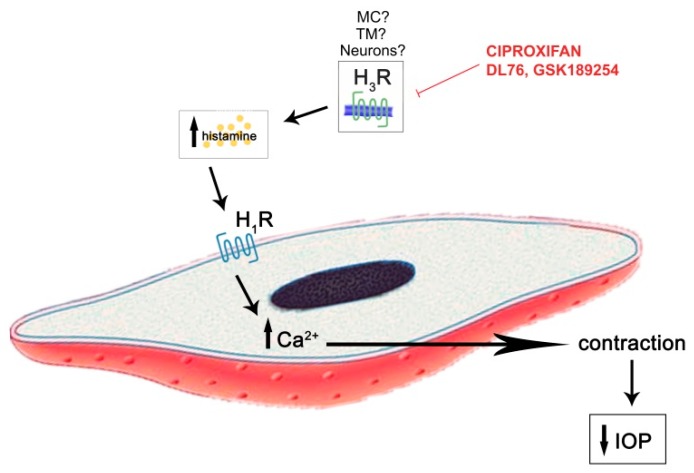
Hypothesized mechanism of action of H_3_ receptor (H_3_R) antagonists in intraocular pressure (IOP) reduction: T-bar meaning antagonism, arrows meaning the hypothetic extracellular and intracellular mechanism of action of H_3_ receptor antagonists, upward and downward arrows meaning increase and decrease respectively.

**Table 1 ijms-20-00981-t001:** Histamine receptor gene sequences used.

Name	Primer Sequence (5′-3′)	TM (°C)	Name	Primer Sequence (5′-3′)	TM (°C)	Product Length (bp)
r H_1_R F ^a^	AGCATGGAACGTCCAGTAGT	58.73	r H_1_R R	TTGCCCTCACACATCCTGTC	59.96	109
r H_2_R F ^b^	GGATACCGCGACTACGAACC	60.32	r H_2_R R	ATCGTGGGAAAGCTGACACG	60.67	94
r H_3_R F ^c^	TCACTGGAGAAGCGCATGAA	59.68	r H_3_R R	GAGCCCAAAGATGCTCACGA	60.39	114
r H_4_R F ^d^	GTAGGAAACGCGGTGGTCAT	60.39	r H_4_R R	TGAGCCAAAACAGGCAGACT	59.82	184

^a^ XM_002713049.2 PREDICTED: Oryctolagus cuniculus histamine receptor H_1_ (*h*H_1_R), transcript variant X1, mRNA, https://www.uniprot.org/uniprot/G1SGD8; ^b^ XM_008255497.1 PREDICTED: Oryctolagus cuniculus histamine H_2_ receptor-like (LOC103348006), mRNA, https://www.uniprot.org/uniprot/P25021; ^c^ NM_001082226.1 Oryctolagus cuniculus histamine receptor H_3_ (*h*H_3_R), mRNA, https://www.uniprot.org/uniprot/Q0Z7Q9; ^d^ XM_002713460.2 PREDICTED: Oryctolagus cuniculus histamine receptor H_4_ (*h*H_4_R), mRNA, https://www.uniprot.org/uniprot/G1TKW6.

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
