# Peer review of "Role of Histamine H3 Receptor Antagonists on Intraocular Pressure Reduction in Rabbit Models of Transient Ocular Hypertension and Glaucoma"

_ijms, 2019, doi:10.3390/ijms20040981_

Reviewer 1 Report

The authors demonstrated that several structurally different H3 antagonists could induce intraocular pressure reduction in two experimental models of glaucoma. They raised the possibility that H3 receptor could be a novel therapeutic target of glaucoma. The studies were well designed and provided with the convincing results. I would like to add some comments on the manuscript to consolidate their hypothesis.
Because the expression levels of β-actin are totally different among various tissues, it is not adequate to compare the relative expression levels of the histamine receptor proteins among different tissues based on those of beta-actin. Although we can find many similar experiments in the published papers, such analyses might not provide significant insights. We could not conclude whether the expression levels of H1 and H4 receptors in retina and optic nerve are higher than those in TM or the expression levels of H3 receptors in retina and optic nerve are lower than those in TM. It depends on the expression levels of β-actin in these tissues. Furthermore, stomach is not a good control tissue in the analysis of H3 receptor protein expression.
The genes of H1 and H2 receptors are intronless and it is possible that the amplified bands contain the genomic DNA-derived products. The authors could verify this possibility by perform the similar experiments without reverse transcription.
To my knowledge, the expression of H3 receptors has never been reported in mast cells. Which kinds of cells might be the targets of H3 receptor antagonists? It is possible that compound 48/80-mediated massive release of histamine should down-modulate the H1 receptors involved in regulation of the intraocular pressure and that neuronal, not mast cell-derived, histamine should play critical roles.
The route of drug administration is important when the readers evaluate the results. Wolf et al. reported that eye drops of an H1 antagonist did not affect the intraocular pressure of glaucoma patients (Eur. J. Ophthalmol. 5, 225-9, 1995). The authors should show the route in the legends to Figures.

Author Response

1. We agree with the observation of the referee and we removed the sentence from the text where we concluded that the expression levels of H1 and H4 receptors in the retina and optic nerve are higher than those in TM or the expression levels of H3 receptors in the retina and optic nerve are lower than those in TM. Stomach tissue was used as a positive control for H2 receptor, as we did not find the signal for this receptor in ocular tissues. Indeed, rising evidence confirm the expression of H3receptors in non-neuronal tissues, stomach included (1, 2).

(1)Panula, P.; Chazot, P.L.; Cowart, M.; Gutzmer, R.; Leurs, R.; Liu, W.L.S.; Stark, H.; Thurmond, R.L.; Haas, H.L. International Union of Basic and Clinical Pharmacology. XCVIII. Histamine Receptors. Pharmacol. Rev. 2015, 67, 601–655.

(2) Grandi D, Shenton FC, Chazot PL, Morini G. Immunolocalization of histamine H3 receptors on endocrine cells in the rat gastrointestinal tract. HistolHistopathol. 2008, 23:789–798

2. We agree with the referee statement, that the amplified bands of H1 and H2 receptors could contain the genomic DNA-derived products because they are intronless. In order to exclude this possibility during the RNA extraction, we treated all the samples with DNAse, previous of the cDNA amplification reactions.

3. We partially agree with this observation on H3R expression on mast cells. The expression of the H3R in mast cell is still debated although some lines of evidence showed H3R expression in LAD2 cell line (1), a well studied human mast cell line (2). The expression of H3R in mast cells was not investigated in our model.  The target of H3R antagonism could be represented by trabecular meshwork cells, however, we cannot exclude an involvement of mast cells. Further studies are needed to address this hypothesis. Our idea is that H1 antagonism or a total absence of histamine could explain the missed effect on IOP reduction. It is reasonable that also the down-regulation of H1 receptor, due to massive histamine release provoked by C48/80, could explain it.

(1) Rudolph MI, Boza Y, Yefi R, Luza S, Andrews E, Penissi A, Garrido P, Rojas IG. The influence of mast cell mediators on migration of SW756 cervical carcinoma cells. J Pharmacol Sci. 2008 Feb;106(2):208-18.

(2) Kirshenbaum AS, Petrik A, Walsh R, Kirby TL, Vepa S, Wangsa D, Ried T, Metcalfe DD. A ten-year retrospective analysis of the distribution, use and phenotypic characteristics of the LAD2 human mast cell line. Int Arch Allergy Immunol. 2014;164(4):265-70. doi: 10.1159/000365729.

4. The route of administration is very important and we added it to the legends of the figures even though it is well explained in the method session paragraph 4.5. We previously demonstrated that the administration of the compound as eye drops produced retinal concentration of the molecule similar to those measured after oral administration with a meaningful difference in plasma concentration (1). Moreover, in the present study we verified the presence of ciproxifan measuring it on aqueous humor after dosing (data not shown). 

(1) Padrini L, Isacchi B, Bilia AR, Pini A, Lanzi C, Masini E, Della Bona ML, Calvani AM, Ceccantini R, la Marca G, Filippi L. Pharmacokinetics and local safety profile of propranolol eye drops in rabbits. Pediatr Res. 2014Oct;76(4):378-85).

Reviewer 2 Report

With interest, I read the manuscript ijms-441622.

Major comments:

1. Page 2 of 14, lines 36-29. The Authors list the major task of the study, of which validation of  H3 receptor antagonists in the modulation of IOP and in baroprotection is only a small part. However, most of the data derive from the experiments related to this aim. I think that while listing the aims and discussing the data the Authors should be refocus on the candidate drugs while consider the other data mostly the introduction to therapeutic investigations. Of course, the other data are also interesting but they are in the absolute minority in this manuscript.

2. Furthermore, the Authors should not ovoid discussion on which tested drug is best and which is worse. A paragraph addressing this issue should be added to the Discussion. For example, Figure 2 clearly shows that GSK189245 is the worst, at least in this setting. Etc. I mean, if this manuscript present drug data, the Reader should benefit through getting to know which of the candidate drugs should be followed in further studies.

3. Page 4 of 14, lines 1-3. It would be good to refer to the relevant papers describing the development or at least the source of the tested drugs, such as e.g. https://www.ncbi.nlm.nih.gov/pubmed/21544189 in case of DL 76.

4. Page 5 of 14, lines 11-14 (“… an effect that could be explained by the release of endogenous histamine.”). Maybe I overlooked something but does it really explain everything? None of the drugs used affects the levels of histamine. Or?

5. Nevertheless, according to what you write, mast cells are the source of the substantial portion of the histamine playing a role in the IOP regulation. In this case, it is tempting to ask if, considering an important  there are (or there could be, in view of your data) any relationships between glaucoma and allergic conjunctivitis, a an eye disease in the pathogenesis of which mast calls and histamine are strongly and directly involved (https://www.ncbi.nlm.nih.gov/pubmed/15637556). Please, speculate in the Discussion section. Existence of such relationship could have some consequences not only for the pathophysiology but also for the treatment.

6. Graphical abstract. I would not use human eye here as you do not do any human experiments. Besides, in my view, the majority of the Readers will not see any big difference between the two eyes; maybe you could draw something, show IOP graphs additionally, etc.? I would more clearly indicate the negative effects of H3 receptor antagonism (there are commonly accepted way of showing an negative/suppressing/blocking/etc. effect.). Would oxidative stress also be blocked by H3 receptor antagonism? If yes, please, show it.

7. Figure 1 A-C and relevant part of the Results, Discussion section, page 7 of 14, lines 19-21. Why H2R mRNA is produced while the protein not? What is behind? What can be the mechanism? Regulatory RNAs (https://www.ncbi.nlm.nih.gov/pubmed/28322581)? Something else?

8. Page 9 of 14. Lines 4-5. Is the “authorization of Italian Ministry of Health (number 1179/2015-PR)” equal to a specific ethics committee agreement for you to conduct this study or is it a general law? Please, clarify

Minor comments:

1. A facultative comment. In experiments shown in e.g. Figures 2 and 3, the comparisons are made between the groups. Did you also compare within the groups, i.e. between time points?

2. Page 8 of 14, lines 15-17. This sentence is unclear. This sentence is unclear. via Gq and phospholipase are downstream of H1 receptor. Please, rebuild.

3. Please, be cared full with details. For example, for some American manufacturers you provide state and country and for some state only. And other mistakes, e.g. page 10 of 14, line 39: no “,” before “Darmstadt”; line 44: “,” not “;” after “Genequant-Pro” and “,” after “Italy”. Some companies do not have any accompanying city, state and country. Etc.

4. It should be “Katarzyna Kieć-Kononowicz” not “Katarzyna Kieć-Kononowicza”, right? This mistake appears twice.

5. Table 1. Pleas,e make it clead what “a” - “d” correspond to (reference sequences). In addition, please, provide www links to all four sequences for the convenience of the Reader.

6. Page 12 of 14, line 11. If “one-way”, then also “two-way”.

Author Response

1. According with the referee’s suggestion, the aim in the text was changed in: “This research’s aims included validating H3 receptor antagonists, both imidazole (ciproxifan) and non-imidazole compounds (DL76 and GSK189254) in IOP reduction, and establishing whether the histamine system plays a role in baroprotection and, consequently, in retinal neuroprotection by reducing oxidative stress. The project also investigated the presence and distribution of histamine receptor subtypes in the retina, ciliary bodies, trabecular meshwork (TM).”

2. We added the paragraph.  “GSK189254 is an H3R antagonist with high affinity for human (pK(i) = 9.59 -9.90) and rat (pK(i) = 8.51-9.17) H3Rs (1,2). No data of rabbit pK(i) is present in scientific literature. NeverthelessGSK189254 was less effective in reducing IOP in our models

(1) Medhurst AD et al. GSK189254, a novel H3 receptor antagonist that binds to histamine H3 receptors in Alzheimer's disease brain and improves cognitive performance in preclinical models. J PharmacolExpTher. 2007 Jun;321(3):1032-45.

(2) Panula, P.; Chazot, P.L.; Cowart, M.; Gutzmer, R.; Leurs, R.; Liu, W.L.S.; Stark, H.; Thurmond, R.L.; Haas, H.L. International Union of Basic and Clinical Pharmacology. XCVIII. Histamine Receptors. Pharmacol. Rev. 2015, 67, 601–655.

3. The source of the tested drugs is explained in the methods section paragraph 4.4. Compounds used in the experiments, and in Table A1. The reference articles about DL76, the only novel H3R antagonist were chosen by the authors who provided the compound (Dr. Dorota Łażewska and Prof KatarzynaKieć-Kononowicz from Department of Technology and Biotechnology of Medicinal Drugs, Jagiellonian University, Kraków, Poland).

References in the manuscript about DL76 are:

- Szafarz, M.; Kryczyk, A.; Łażewska, D.; Kieć-Kononowicz, K.; Wyska, E. Pharmacokinetics and tissue distribution of the new non-imidazole histamine H3 receptor antagonist 1-[3-(4- tert -butylphenoxy) propyl]piperidine in rats. Xenobiotica. 2015, 45, 912–920. ht34.

- Łażewska, D.; Ligneau, X.; Schwartz, J.-C.; Schunack, W.; Stark, H.; Kieć-Kononowicz, K. Ether derivatives of 3-piperidinopropan-1-ol as non-imidazole histamine H3 receptor antagonists. Bioorg. Med. Chem. 2006, 14, 3522–3529. https://doi.org/10.1016/j.bmc.2006.01.013

-Łażewska, D.; Kaleta, M.; Schwed, J.S.; Karcz, T.; Mogilski, S.; Latacz, G.; Olejarz, A.; Siwek, A.; Kubacka, M.; Lubelska, A.; et al. Biphenyloxy-alkyl-piperidine and azepane derivatives as histamine H3 receptor ligands. Bioorg. Med. Chem. 2017, 25, 5341–5354. doi: 10.1016/j.bmc.2017.07.058. https://dx.doi.org/10.3109/00498254.2015.1025117

We added on the text the references suggested

- Szafarz M, Szymura-Oleksiak J, Lazewska D, Kiec-Kononowicz K. LC-MS-MS Method for the Analysis of New Non-Imidazole Histamine H(3) Receptor Antagonist 1-[3-(4-tert-Butylphenoxy)propyl]piperidine in Rat Serum-Application to Pharmacokinetic Studies. Chromatographia. 2011, 73, 913-919. https://dx.doi.org/10.1007/s10337-011-1983-9

As for the other compounds we referred to a very important pharmacological review that describes the characteristics of all the compounds used for our studies.

- Panula, P.; Chazot, P.L.; Cowart, M.; Gutzmer, R.; Leurs, R.; Liu, W.L.S.; Stark, H.; Thurmond, R.L.; Haas, H.L. International Union of Basic and Clinical Pharmacology. XCVIII. Histamine Receptors. Pharmacol. Rev. 2015, 67, 601–655.

4. We agree with the referee’s observation it could be interesting to measure the levels of endogenous histamine for example in aqueous humor. Unfortunately, we did not perform the analysis also in consideration of the fact that probably a small quantity of histamine is released and it could be rapidly metabolized and difficult to be measured. In agreement with your comment we deleted the sentence in the text because it was not pertinent to that paragraph on specificity of the compounds used. We added a new sentence on this specific subject in the discussion: “It is possible that compound 48/80-mediated massive release of histamine could down-modulate the H1 receptors involved in regulation of the intraocular pressure reproducing the effect of h1 blockade observed with pyrilamine pretreatment.”

5. In the transient IOP model mast cells could work as histamine providing cells in a paracrine way under stimuli of H3 antagonist directly (we don’t know yet)or via another mediator.  Human mast cells express all the histamine receptor subtypes (1).

To our knowledge, a clear and direct relationship between conjunctivitis and glaucoma has not been established.  It has been demonstrated that repeated administration of topical corticosteroids is clearly associated with an increased risk for the development of ocular hypertension and glaucoma (2, 3), probably due to a heavy modification of the trabecular meshwork ultrastructure, rather than an histamine machinery involvement or mast cells activation. Furthermore, we can speculate that mast cell massive degranulation, induced by C48/80, could account for H1R down-regulation.

(1) Thangam EB, Jemima EA, Singh H, et al. The Role of Histamine and Histamine Receptors in Mast Cell-Mediated Allergy and Inflammation: The Hunt for New Therapeutic Targets. Front Immunol. 2018;9:1873. Published 2018 Aug 13. doi:10.3389/fimmu.2018.01873

(2) Whitlock NA, McKnight B, Corcoran KN, Rodriguez LA, Rice DS. Increased intraocular pressure in mice treated with dexamethasone. Invest Ophthalmol Vis Sci. 2010 Dec;51(12):6496-503.

(3)Jones R 3rd, Rhee DJ. Corticosteroid-induced ocular hypertension and glaucoma: a brief review and update of the literature. CurrOpinOphthalmol. 2006 Apr;17(2):163-7. Review

6. We restyled the graphical abstract with a cartoon representing the normal and the hypertensive eye.   We could not exclude that this effect of oxidative stress reduction might be baroprotection mediated.

7.The levels of mRNA are not reliable indicators for corresponding protein levels without verification.In fact, microRNAs are known to repress hundreds of genes by inhibiting mRNA translation into protein and so modulate a great variety of mammalian cellular processes (1).Many post-translational mechanisms important for the control of protein turnover have been described; for example, protein levels are affected by activation of ligases and proteases that regulate protein degradation,and post-translational phosphorylation, that control cell division and differentiation, (2). Finally, endocytosis-lysosome system are known to control protein half-lives. Considering all the ways in which protein levels are controlled post-translationally, and given that mRNA translation is regulated by microRNAs in many cases, the correlation between mRNAand protein levels is not directly proportional.

(1) Bartel, D.P., MicroRNAs: target recognition and regulatory functions. Cell, 2009. 136(2): p. 215-33.

(2) Landry, J.J., P.T. Pyl, T. Rausch, T. Zichner, M.M. Tekkedil, A.M. Stutz, A. Jauch, R.S. Aiyar, G. Pau, N. Delhomme, J. Gagneur, J.O. Korbel, W. Huber, and L.M. Steinmetz, The genomic and transcriptomic landscape of a HeLa cell line. G3 (Bethesda), 2013. 3(8): p. 1213-24. 

8. It is a specific Ethics Committee agreement approved by both a local Ethics Committee and by a special Ethics Committee of the Ministry of Health for all the projects involving laboratory animals

minor comments 

 We did not compare the IOP change within the groups at each timepoint.The IOP change was calculated as the difference between the mean value of IOP in the different treatments versus the mean IOP in vehicles at the corresponding timepoints. 

We rebuilded the sentence ”Histamine could interact with H1 receptor, aG-protein receptor (Gq), on the cell surface and activates phospholipase C and the phosphatidylinositol signal pathway, thus increasing intracellular calcium and subsequently inducing cell contraction. Histamine, via H1R, stimulates the production of inositol phosphate and mobilization of intracellular calcium2+ in cultured human ciliary muscle cells reducing IOP [9,16]”.

Checked 

KatarzynaKieć-Kononowicz is the right spelling

Table 1 was restyled

We checked and corrected it in figure 5 legend

Reviewer 3 Report

The authors investigated the presence and the different distribution of the histamine receptor subtypes in the retina, ciliary bodies, trabecular meshwork (TM) so as to validate H3 receptor antagonists in the modulation of IOP and in baroprotection. The authors concluded that H3R antagonists could represent a future promising therapy for glaucoma. The findings are interesting and may provide a novel therapeutic strategy for glaucoma. It might be better for readers if the authors could address a couple of points described below.

1. One might consider how imetit, pyrilamine or compound 48/80 alone might affect IOP.

2. It might be helpful for readers if a schematic diagram could be provided to show the underlying mechanisms responsible for modulation of IOP and baroprotection of H3R antagonists, including the possible target tissues (retinopetal histaminergic neurons, ciliary bodies, TM, RGCs, mast cells, etc.) and receptor subtypes present. 

Author Response

1. We also tested the drugs alone and they didn’t affect IOP in normotensive rabbits (data not shown).  

2. According to the reviewer comment, we added a schematic diagram (Figure 6 in the Discussion Section) hypothesizing the IOP modulation mechanism elicited by H3R antagonists. We are not able to speculate in the diagram on the role of each different cell type in relation to H3R antagonism becausedata are not yet available on the contribution/involvement of single cell type.

Human mast cell express all the 4 histamine receptor subtypes (1). LAD2 cells are a human mast cell line expressing H3receptors  (2).

(1) Thangam EB, Jemima EA, Singh H, et al. The Role of Histamine and Histamine Receptors in Mast Cell-Mediated Allergy and Inflammation: The Hunt for New Therapeutic Targets. Front Immunol. 2018;9:1873. Published 2018 Aug 13. doi:10.3389/fimmu.2018.0187

(2) Rudolph MI, Boza Y, Yefi R, Luza S, Andrews E, Penissi A, et al. The influence of mast cell mediators on migration of SW756 cervical carcinoma cells. J PharmacolSci (2008) 106:208–18.10.1254/jphs.FP0070736

English language and style were checked by  a mother tongue translator 

Round  2

Reviewer 1 Report

The authors adequately addressed the concerns. I have no specific comments.